# Early predictive factors of progression from severe type to critical ill type in patients with Coronavirus Disease 2019: A retrospective cohort study

**Nan Li[1][☯], Hao Kong[1][☯], Xi-Zi Zheng[2], Xue-Ying Li[3], Jing Ma[4], Hong Zhang[4], Dong-Xin Wang**  **[1]\*, Hai-Chao Li[4]\*, Xin-Min Liu[5]**

1 Department of Anesthesiology and Critical Care Medicine, Peking University First Hospital, Beijing, China, 2 Department of Nephrology, Peking University First Hospital, Beijing, China, 3 Department of Biostatistics, Peking University First Hospital, Beijing, China, 4 Department of Respiratory and Critical Care Medicine, Peking University First Hospital, Beijing, China, 5 Department of Geriatrics, Peking University First Hospital, Beijing, China

☯ These authors contributed equally to this work.
\* dxwang65@bjmu.edu.cn, wangdongxin@hotmail.com (DXW); lhch91767@sina.com (HCL)

**Data Availability Statement:** All relevant data are within the manuscript and its Supporting Information files.

## Abstract

### Background

The current worldwide pandemic of Coronavirus Disease 2019 (COVID-19) has posed a serious threat to global public health, and the mortality rate of critical ill patients remains high. The purpose of this study was to identify factors that early predict the progression of COVID-19 from severe to critical illness.

### Methods

This retrospective cohort study included adult patients with severe or critical ill COVID-19 who were consecutively admitted to the Zhongfaxincheng campus of Tongji Hospital (Wuhan, China) from February 8 to 18, 2020. Baseline variables, data at hospital admission and during hospital stay, as well as clinical outcomes were collected from electronic medical records system. The primary endpoint was the development of critical illness. A multivariable logistic regression model was used to identify independent factors that were associated with the progression from severe to critical illness.

### Results

A total of 138 patients were included in the analysis; of them 119 were diagnosed as severe cases and 16 as critical ill cases at hospital admission. During hospital stay, 19 more severe cases progressed to critical illness. For all enrolled patients, longer duration from diagnosis to admission (odds ratio [OR] 1.108, 95% CI 1.022–1.202; P = 0.013), pulse oxygen saturation at admission <93% (OR 5.775, 95% CI 1.257–26.535; P = 0.024), higher neutrophil count (OR 1.495, 95% CI 1.177–1.899; P = 0.001) and higher creatine kinase-MB level at admission (OR 2.449, 95% CI 1.089–5.511; P = 0.030) were associated with a higher risk, whereas higher lymphocyte count at admission (OR 0.149, 95% CI 0.026–0.852; P = 0.032)

**Funding:** The authors received no specific funding for this work.

**Competing interests:** The authors have declared that no competing interests exist.

was associated with a lower risk of critical illness development. For the subgroup of severe cases at hospital admission, the above factors except creatine kinase-MB level were also found to have similar correlation with critical illness development.

### Conclusions

Higher neutrophil count and lower lymphocyte count at admission were early independent predictors of progression to critical illness in severe COVID-19 patients.

## Introduction

The current worldwide pandemic of coronavirus disease 2019 (COVID-19) caused by the severe acute respiratory syndrome coronavirus 2 (SARS-CoV-2) has posed a serious threat to global public health. As of October 14, 2020, nearly 40 million confirmed cases and more than 1 million deaths have been reported in over 2 hundred countries. In China, a nationwide study indicated the proportion of severe/critical cases was around 7–10% [1] and the mortality was about 4.1% according to the information released by the Chinese National Health Commission [2].

The severity of COVID-19 varies from mild, moderate, severe to critical ill types according to the Guideline for Diagnosis and Treatment of Novel Coronavirus Infection (Trial Version 5) [3]. Patients with older age, chronic smoking, comorbidities (hypertension, diabetes mellitus, chronic obstructive pulmonary disease, and cardiovascular disease) and cancer are at higher risk of poor prognosis and even death [4–8]. And mortality increases with the progression of severity. The study of Guan et al. [4] included 1,099 patients with laboratory-confirmed COVID-19 from 552 hospitals in 31 provinces of China, and showed that the mortality rate of non-severe (mild and moderate type) and severe (severe and critical ill type) patients were 0.1% and 8.1%, respectively. Yang et al. [5] observed 52 critically ill COVID-19 patients, defined as requirement of mechanical ventilation or a fraction of inspired oxygen of 60% or more, and reported a mortality rate of 61.5%. Two case series from the United States also reported similar mortality rate of 50% and 67%, respectively, in critically ill COVID-19 patients [9,10].

At present, there are no specific drugs or vaccines for COVID-19; the mainstay of treatment is supportive care. Effective interventions that can slow down or prevent disease progression from non-severe to severe or from severe to critically ill are the key of saving life. Therefore, it is important to identify factors that can early predict the progression of COVID-19. In a previous study of Wu et al. [11], old age, comorbidities and late initiation of antiviral treatment were associated with higher risk of COVID-19 progression. However, few studies investigated the predictors of progression from severe to critical illness. In the clinical scenario, patients admitted to hospital with severe symptoms may progress to critical illness within a few days or hours, leaving limited time to deal with. The purpose of this study was to identify factors that early predict the progression of COVID-19 from severe to critical illness.

## Materials and methods

### Study design and participants

This retrospective cohort study involved patients who were consecutively admitted to the Zhongfaxincheng campus of Tongji Hospital (Wuhan, China) from February 8 to 18, 2020. This campus was designated for severe and critically ill patients and was taken over by a

medical team from Peking University. The study protocol was approved by the Clinical Research Ethics Committee of Peking University First Hospital, Beijing, China (2020 [077]) on March 13, 2020.

The inclusion criteria were: (1) age ≥18 years; (2) laboratory-confirmed COVID-19; and (3) severe or critically ill cases. The exclusion criteria were patients with missing data of primary endpoint, i.e., development of critical illness or not during hospital stay. Considering that the study was retrospective in nature and no follow-up was performed, the Ethics Committee agreed to waive written informed consent. Personal data of patients were kept strictly confidential. This manuscript adheres to the applicable STROBE guidelines.

## Clinical management

All patients received nasopharyngeal swab sampling and were tested by real-time reverse transcriptase-polymerase chain reaction assays before admission. The diagnosis of COVID-19 was confirmed according to the Guidelines for the Diagnosis and Treatment of Novel Coronavirus Infection [3] and the World Health Organization interim guidance [12]. At hospital admission, symptoms, comorbidities and pre-hospital treatments were obtained from self-report or by asking family members via telephone. Vital signs were recorded. Full blood count, biochemical tests, and coagulation assays were performed and evaluated for all patients.

As a routine practice, all severe patients were provided with oxygen therapy. For those whose pulse oxygen saturation was 93% or less while breathing ambient air and/or respiratory rate was 30 breaths per minute or higher, an initial oxygen therapy with a flow of 5 L/min was started. The flow of oxygen was adjusted and the oxygen delivery systems (nasal prong, oxygen mask, or non-rebreathing mask) were selected according to the severity of hypoxia. For patients who were given non-rebreathing mask with an oxygen flow of 10–15 L/min but still had a pulse oxygen saturation of 90% or less or a respiratory rate of 30 breaths per minute or higher, i.e., those with suspected severe acute hypoxic respiratory failure or acute respiratory distress syndrome, mechanical ventilation was initiated. Antiviral drugs therapy was provided according to physicians' discretion. Antibiotics were added when bacterial infection was highly suspected or confirmed. Gamma globulin and glucocorticoids were administered in some critically ill cases.

## Definition of severe and critically ill COVID-19

Severe and critically ill COVID-19 were diagnosed according to the 5th version Chinese Guidelines for the Diagnosis and Treatment of Novel Coronavirus Infection [3]. Severe COVID-19 was defined as having any of the following: (1) respiratory distress with respiratory rate ≥30 breaths per minute; (2) pulse oxygen saturation of ≤93% in resting-state; or (3) $PaO_2/FiO_2$ ≤300 mmHg. Critically ill COVID-19 was defined as severe cases having any of the following: (1) respiratory failure requiring mechanical ventilation; (2) shock; or (3) dysfunction of other organs.

## Data collection

The data were extracted from electronic medical records and reviewed by a trained team of physicians. Demographics (age and sex), symptoms since onset, comorbidities, smoking history, time from onset to hospital admission, time from diagnosis to hospital admission, treatments before admission (antiviral treatment, antibiotics, nonsteroidal anti-inflammatory drugs, glucocorticoids, and gamma globulin), vital signs, laboratory tests at admission (full blood count, biochemical tests, coagulation assays), treatments after hospital admission (therapeutic medications and respiratory supports), and clinical outcomes were collected. The

primary endpoint was the development of critical illness, either at hospital admission or during hospital stay.

## Statistical analysis

All enrolled patients were divided into two groups according to the development of critical illness. The normality of data was tested using the Kolmogorov–Smirnov test. Continuous variables were compared using the student's t-test (normal distribution) or Mann–Whitney U-test (non-normal distribution). Categorical variables were analyzed using the chi-squared test or Fisher's exact test.

Univariable logistic regression analyses were performed to screen factors that might be associated with the development critical illness. For factors with P ≤0.10, the presence of collinearity was determined using Pearson correlation test or Spearman correlation test; for those with collinearity or clinical relevance, only one parameter was included for further analysis. A multivariable logistic regression model was used to identify independent factors that were associated with the progression from severe type to critically ill type with Wald (backward) method. A P value of <0.05 (two-sided) was considered statistically significant. All statistical analyses were performed using SPSS software for Windows (version 22.0; SPSS, Inc., Chicago, IL, USA).

## Results

### Patients

Between February 8 to 18, 2020, 138 patients with diagnosed COVID-19 were consecutively admitted. All patients met the inclusive/exclusion criteria and were included in the analysis. Among the enrolled patients, 119 were diagnoses as severe cases and 16 as critically ill cases at hospital admission; during hospital stay, 19 more severe cases progressed to critical illness, resulting a total of 35 critically ill cases (Fig 1).

Of all enrolled patients, the mean age was 62 (SD 14) years, and 51.4% (71/138) were male. Symptoms that appeared in more than half of patients included fever (86.2%), cough (84.1%), dyspnea (66·7%), and expectoration (62.0%). The median time from symptom onset to hospital admission was 14 (interquartile range [IQR] 11–18) days. The median time from diagnosis to hospital admission was 8 (IQR 4–13) days. Before admission, 71.0% (98/138) of patients received antiviral drugs and 68.1% (94/138) received antibiotics (Table 1).

### Baseline characteristics and variables at admission

Of all enrolled patients, when compared with severe cases without aggravation, those who developed critical illness were older, and received fewer antiviral drugs (especially arbidol) and antibiotics (especially fluoroquinolones) before admission (Table 1). At hospital admission, patients who developed critical illness had a faster respiratory rate, a greater proportion with pulse oxygen saturation <93%, higher white blood cell and neutrophil counts, but a lower lymphocyte count; regarding biochemical test results, they had higher serum levels of aspartate aminotransferase, creatinine, blood urea nitrogen, lactate dehydrogenase, myoglobin, hypersensitive cardiac troponin I, creatine kinase-MB, and N-terminal pro-brain natriuretic peptide, but a lower level of serum albumin; regarding coagulation, they had a longer prothrombin time and a higher D-dimer level (Table 2; S1 Table).

In the subgroup of severe cases at hospital admission, when compared with cases without aggravation, those who developed critical illness received fewer arbidol before admission (Table 1). At hospital admission, patients who developed critical illness had a faster respiratory

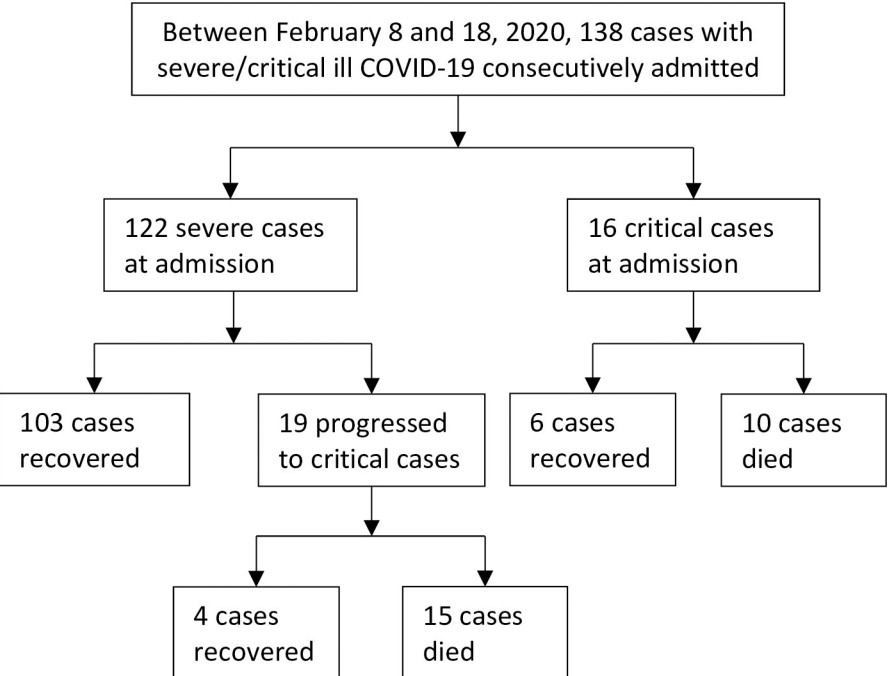

**Fig 1. Study flowchart.** COVID-19, Coronavirus Disease 2019.

rate, a greater proportion with pulse oxygen saturation <93%, higher white blood cell and neutrophil counts, but a lower lymphocyte count; regarding biochemical test results, they had higher serum levels of aspartate aminotransferase, blood urea nitrogen, lactate dehydrogenase, myoglobin, hypersensitive cardiac troponin I, creatine kinase-MB, and N-terminal pro-brain natriuretic peptide; regarding coagulation, they had a longer prothrombin time, a longer activated partial thromboplastin time, and a higher D-dimer level (Table 2; S1 Table).

## Treatments after admission and outcomes

Of all enrolled patients, when compared with severe cases without aggravation, those who developed critical illness received more antibiotics (especially fluoroquinolones), more glucocorticoids, more gamma globulin, and more noninvasive/invasive ventilation; they developed critical illness in a median 2 (IQR 0–8) days, and had a higher in-hospital mortality. Those who survived had a longer hospital stay (Table 3).

In the subgroup of severe cases at hospital admission, when compared with cases without aggravation, those who developed critical illness received more antibiotics, more glucocorticoids, and more noninvasive/invasive ventilation; they developed critical illness in a median 5 (IQR 4–11) days and had a higher in-hospital mortality (Table 3).

## Predictors of progression from severe type to critically ill type

Of all enrolled patients, univariable analysis identified 26 factors with P ≤0.10 (S2 Table). After excluding factors having collinearity or clinical relation with others, 13 factors were included in the multivariate logistic regression model. Five factors were identified to be independently associated with the development of critical illness in COVID-19 patients; of them longer duration from diagnosis to admission (odds ratio [OR] 1.108, 95% CI 1.022–1.202; P = 0.013), pulse oxygen saturation at admission <93% (OR 5.775, 95% CI 1.257–26.535;

**Table 1. Baseline characteristics.**

| Variables | All (n = 138) | All patients | | | Severe cases at admission | | |
|---|---|---|---|---|---|---|---|
| | | Severe cases (n = 103) | Critical cases (n = 35) | P value | Severe cases (n = 103) | Critical cases [a] (n = 19) | P value |
| Demographics | | | | | | | |
| Age, years | 62±14 | 61±15 | 67±12 | **0.025** | 61±15 | 66±15 | 0.159 |
| Male gender | 71 (51.4%) | 48 (46.6%) | 23 (65.7%) | 0.051 | 48 (46.6%) | 13 (68.4%) | 0.081 |
| Symptoms since onset | | | | | | | |
| Fever | 119 (86.2%) | 89 (86.4%) | 30 (85.7%) | >0.999 | 89 (86.4%) | 17 (89.5%) | >0.999 |
| Peak body temperature, °C | 38.6±0.7 | 38.6±0.8 | 38.6±0.7 | 0.925 | 38.6±0.8 | 38.7±0.7 | 0.630 |
| Cough | 116 (84.1%) | 85 (82.5%) | 31 (88.6%) | 0.398 | 85 (82.5%) | 17 (89.5%) | 0.736 |
| Dyspnea | 92 (66.7%) | 64 (62.1%) | 28 (80.0%) | 0.053 | 64 (62.1%) | 14 (73.7%) | 0.335 |
| Expectoration | 85 (62.0%) | 59 (57.8%) | 26 (74.3%) | 0.084 | 59 (57.8%) | 15 (78.9%) | 0.083 |
| Myalgia | 66 (48.2%) | 49 (48.0%) | 17 (48.6%) | 0.957 | 49 (48.0%) | 10 (52.6%) | 0.713 |
| Headache | 49 (35.5%) | 35 (34.0%) | 14 (40.0%) | 0.520 | 35 (34.0%) | 6 (31.6%) | 0.839 |
| Nausea | 47 (34.1%) | 37 (35.9%) | 10 (28.6%) | 0.428 | 37 (35.9%) | 6 (31.6%) | 0.716 |
| Palpitation | 36 (26.3%) | 25 (24.5%) | 11 (31.4%) | 0.422 | 25 (24.5%) | 6 (31.6%) | 0.570 |
| Night sweats | 32 (23.4%) | 25 (24.5%) | 7 (20.0%) | 0.586 | 25 (24.5%) | 4 (21.1%) | >0.999 |
| Sore throat | 30 (21.7%) | 25 (24.3%) | 5 (14.3%) | 0.216 | 25 (24.3%) | 4 (21.1%) | >0.999 |
| Vomiting | 29 (21.0%) | 22 (21.4%) | 7 (20.0%) | 0.865 | 22 (21.4%) | 3 (15.8%) | 0.761 |
| Chest pain | 26 (18.8%) | 19 (18.4%) | 7 (20.0%) | 0.839 | 19 (18.4%) | 3 (15.8%) | >0.999 |
| Hemoptysis | 18 (13.0%) | 11 (10.7%) | 7 (5.1%) | 0.243 | 11 (10.7%) | 3 (15.8%) | 0.456 |
| Comorbidities | | | | | | | |
| Hypertension | 63 (45.7%) | 46 (44.7%) | 17 (48.6%) | 0.688 | 46 (44.7%) | 7 (36.8%) | 0.528 |
| Diabetes | 28 (20.3%) | 20 (19.4%) | 8 (22.9%) | 0.662 | 20 (19.4%) | 7 (36.8%) | 0.130 |
| Coronary artery disease | 26 (18.8%) | 21 (20.4%) | 5 (14.3%) | 0.425 | 21 (20.4%) | 3 (15.8%) | 0.763 |
| Pulmonary diseases [b] | 21 (15.2%) | 13 (12.6%) | 8 (22.9%) | 0.145 | 13 (12.6%) | 3 (15.8%) | 0.714 |
| Chronic kidney diseases [c] | 9 (6.5%) | 5 (4.9%) | 4 (11.4%) | 0.231 | 5 (4.9%) | 2 (10.5%) | 0.299 |
| Smoking history | 29 (22.3%) | 18 (18.4%) | 11 (34.4%) | 0.059 | 18 (18.4%) | 5 (29.4%) | 0.328 |
| From onset to admission, days | 14 (11, 18) | 15 (11, 18) | 13 (9, 17) | 0.453 | 15 (11, 18) | 11 (8, 23) | 0.239 |
| From diagnosis to admission, days | 8 (4, 13) | 8 (5, 13) | 9 (4, 14) | 0.672 | 8 (5, 13) | 8 (3, 26) | 0.942 |
| Treatments before admission | | | | | | | |
| Antiviral drugs [d] | 98 (71.0%) | 79 (76.7%) | 19 (54.3%) | **0.012** | 79 (76.7%) | 12 (63.2%) | 0.253 |
| Arbidol | 38 (27.7%) | 35 (34.0%) | 3 (8.8%) | **0.004** | 35 (34.0%) | 2 (10.5%) | **0.041** |
| Oseltamivir | 46 (33.6%) | 37 (35.9%) | 9 (26.5%) | 0.312 | 37 (35.9%) | 6 (31.6%) | 0.716 |
| Antibiotics | 94 (68.1%) | 75 (72.8%) | 19 (54.3%) | **0.042** | 75 (72.8%) | 11 (57.9%) | 0.190 |
| Fluoroquinolones | 67 (48.6%) | 56 (54.4%) | 11 (31.4%) | **0.019** | 56 (54.4%) | 8 (42.1%) | 0.325 |
| β-lactams | 22 (15.9%) | 15 (14.6%) | 7 (20.0%) | 0.448 | 15 (14.6%) | 3 (15.8%) | >0.999 |
| Macrolides | 4 (2.9%) | 3 (2.9%) | 1 (2.9%) | >0.999 | 3 (2.9%) | 0 (0.0%) | >0.999 |
| Nonsteroidal anti-inflammatory drugs | 16 (11.6%) | 14 (13.6%) | 2 (5.7%) | 0.358 | 14 (13.6%) | 2 (10.5%) | >0.999 |
| Glucocorticoids | 16 (11.6%) | 11 (10.7%) | 5 (14.3%) | 0.552 | 11 (10.7%) | 3 (15.8%) | 0.456 |
| Gamma globulin | 12 (8.9%) | 8 (7.9%) | 4 (11.8%) | 0.497 | 8 (7.9%) | 2 (11.1%) | 0.647 |

Data are presented as mean ± SD, number (%), or median (interquartile range).

[a] Patients progressed from severe type to critical ill type.

[b] Includes asthma, chronic obstructive pulmonary disease, and interstitial lung disease.

[c] Defined as glomerular filtration rate (GFR) <60 mL/min per 1·73 m$^2$ or markers of kidney damage, or both, of at least 3 months duration.

[d] Includes arbidol, oseltamivir, lopinavir/ritonavir, interferon, ganciclovir, and ribavirin.

**Table 2. Variables at admission.**

| Variables | All (n = 138) | All patients | | | Severe cases at admission | | |
|---|---|---|---|---|---|---|---|
| | | Severe cases (n = 103) | Critical cases (n = 35) | P value | Severe cases (n = 103) | Critical cases [a] (n = 19) | P value |
| Vital signs | | | | | | | |
| Heart rate, bpm | 98±18 | 96±16 | 103±24 | 0.096 | 96±16 | 101±19 | 0.211 |
| Systolic BP, mmHg | 134±22 | 134±22 | 134±24 | 0.914 | 134±22 | 128±26 | 0.330 |
| Diastolic BP, mmHg | 83±15 | 83±13 | 83±19 | 0.988 | 83±13 | 80±21 | 0.661 |
| Respiratory rate, bpm | 22 (20, 26) | 22 (20, 24) | 26 (23, 32) | **<0.001** | 22 (20, 24) | 26 (24, 30) | **0.003** |
| Pulse oxygen saturation <93% | 48 (35.0%) | 23 (22.5%) | 25 (71.4%) | **<0.001** | 23 (22.5%) | 14 (73.7%) | **<0.001** |
| Full blood count | | | | | | | |
| White blood cell, ×10⁹/L | 5.5 (4.4, 7.7) | 5.1 (4.2, 6.3) | 8.4 (6.7, 13.4) | **<0.001** | 5.1 (4.2, 6.3) | 7.8 (5.1, 11.5) | **0.001** |
| Neutrophil, ×10⁹/L | 4.0 (2.7, 6.0) | 3.4 (2.5, 4.6) | 7.2 (5.1, 12.3) | **<0.001** | 3.4 (2.5, 4.6) | 6.0 (4.3, 8.9) | **<0.001** |
| Lymphocyte, ×10⁹/L | 0.9 (0.6, 1.4) | 1.1 (0.7, 1.5) | 0.7 (0.4, 1.1) | **<0.001** | 1.1 (0.7, 1.5) | 0.9 (0.4, 1.2) | **0.046** |
| Hemoglobin, g/dL | 12.4±2.1 | 12.4±1.3 | 12.7±3.5 | 0.619 | 12.4±1.3 | 12.5±3.4 | 0.831 |
| Platelet, ×10⁹/L | 229±98 | 234±87 | 215±124 | 0.431 | 234±87 | 207±128 | 0.269 |
| Biochemical tests | | | | | | | |
| ALT, U/L | 22 (16, 40) | 22 (14, 40) | 30 (19, 43) | 0.104 | 22 (14, 40) | 30 (18, 43) | 0.211 |
| AST, U/L | 28 (18, 41) | 24 (18, 35) | 40 (34, 53) | **<0.001** | 24 (18, 35) | 41 (34, 53) | **0.001** |
| Albumin, g/L | 34.3±4.8 | 35.1±4.9 | 32.0±3.9 | **0.001** | 35.1±4.9 | 33.2±4.2 | 0.110 |
| Creatinine, μmol/L | 74 (58, 91) | 70 (57, 87) | 86 (64, 105) | **0.006** | 70 (57, 87) | 83 (64, 99) | 0.088 |
| Blood urea nitrogen, mmol/L | 4.7 (3.3, 6.6) | 4.1 (3.0, 5.2) | 8.2 (5.5, 11.8) | **<0.001** | 4.1 (3.0, 5.2) | 7.5 (5.0, 10.8) | **<0.001** |
| Lactate dehydrogenase, U/L | 290 (234, 407) | 267 (227, 328) | 466 (334, 674) | **<0.001** | 267 (227, 328) | 351 (302, 490) | **0.001** |
| Myoglobin, ng/mL | 60.2 (37.9, 131.3) | 48.8 (33.2, 86.6) | 131.8 (76.8, 259.7) | **<0.001** | 48.8 (33.2, 86.6) | 105.9 (58.4, 230.2) | **0.003** |
| Hypersensitive cTnI, pg/mL | 4.8 (2.2, 11.1) | 3.8 (1.9, 7.3) | 19.1 (5.7, 119.4) | **<0.001** | 3.8 (1.9, 7.3) | 11.2 (6.1, 28.7) | **0.001** |
| Creatine kinase-MB, ng/mL | 0.9 (0.4, 1.7) | 0.7 (0.4, 1.3) | 2.0 (1.1, 5.0) | **<0.001** | 0.7 (0.4, 1.3) | 1.1 (0.9, 2.4) | **0.012** |
| NT-proBNP, pg/mL | 185 (67, 468) | 135 (62, 291) | 743 (193, 1498) | **<0.001** | 135 (62, 291) | 483 (156, 995) | **0.001** |
| Coagulation function | | | | | | | |
| Prothrombin time, s | 14.1 (13.5, 14.7) | 13.9 (13.4, 14.4) | 15.3 (14.0, 16.1) | **<0.001** | 13.9 (13.4, 14.4) | 14.6 (13.8, 15.6) | **0.012** |
| APTT, s | 40.4 (36.3, 45.0) | 40.2 (35.7, 44.3) | 41.3 (37.7, 46.2) | 0.053 | 40.2 (35.7, 44.3) | 45.4 (39.6, 51.1) | **0.015** |
| D-dimer, μg/mL | 1.3 (0.5, 2.5) | 0.8 (0.5, 1.9) | 2.7 (1.6, 12.1) | **<0.001** | 0.8 (0.5, 1.9) | 2.0 (1.0, 3.0) | **0.004** |

Data are presented as mean ± SD, number of patients (%), or median (interquartile range).

BP, blood pressure; AST, aspartate aminotransferase; ALT, alanine aminotransferase; cTnI, cardiac troponin I; NT-proBNP, N-terminal pro-brain natriuretic peptide; APTT, activated partial thromboplastin time.

[a] Patients progressed from severe type to critical ill type.

P = 0.024), higher neutrophil count (OR 1.495, 95% CI 1.177–1.899; P = 0.001) and higher creatine kinase-MB level (OR 2.449, 95% CI 1.089–5.511; P = 0.030) at admission were associated with a higher risk, whereas higher lymphocyte count at admission (OR 0.149, 95% CI 0.026–0.852; P = 0.032) was associated with a lower risk of critical illness development (Table 4).

In the subgroup of severe cases at hospital admission, univariable analysis identified 16 factors with P ≤0.10 (S2 Table). After excluding factors having collinearity or clinical relation with others, 11 factors were included in the multivariate logistic regression model. Four factors were identified to be independently associated with the development of critical illness in COVID-19 patients; of them longer duration from diagnosis to admission (OR 1.085, 95% CI 1.009–1.167; P = 0.027), pulse oxygen saturation at admission <93% (OR 11.182, 95% CI 2.426–51.534; P = 0.002) and higher neutrophil count at admission (OR 1.403, 95% CI 1.117–

**Table 3. Treatments after admission and outcomes.**

| Variables | All (n = 138) | All patients | | | Severe cases at admission | | |
|---|---|---|---|---|---|---|---|
| | | Severe cases (n = 103) | Critical cases (n = 35) | P value | Severe cases (n = 103) | Critical cases (n = 19) [a] | P value |
| **Treatments after admission** | | | | | | | |
| Antiviral drugs | 120 (87.0%) | 90 (87.4%) | 30 (85.7%) | 0.777 | 90 (87.4%) | 18 (94.7%) | 0.694 |
| Arbidol | 87 (64.0%) | 67 (65.7%) | 20 (58.8%) | 0.470 | 67 (65.7%) | 13 (68.4%) | 0.817 |
| Lopinavir/Ritonavir | 25 (18.2%) | 18 (17.5%) | 7 (20.6%) | 0.684 | 18 (17.5%) | 3 (15.8%) | >0.999 |
| Oseltamivir | 4 (2.9%) | 3 (2.9%) | 1 (2.9%) | >0.999 | 3 (2.9%) | 1 (5.3%) | 0.497 |
| Antibiotics [b] | 69 (50.0%) | 36 (35.0%) | 33 (94.3%) | **<0.001** | 36 (35.0%) | 18 (94.7%) | **<0.001** |
| Fluoroquinolones | 53 (38.7%) | 34 (33.0%) | 19 (55.9%) | **0.018** | 34 (33.0%) | 10 (52.6%) | 0.102 |
| Glucocorticoids | 34 (24.6%) | 10 (9.7%) | 24 (68.8%) | **<0.001** | 10 (9.7%) | 12 (63.2%) | **<0.001** |
| Gamma globulin | 26 (18.8%) | 14 (13.6%) | 12 (34.3%) | **0.007** | 14 (13.6%) | 6 (31.6%) | 0.085 |
| Respiratory support | 35 (25.4%) | 0 (0.0%) | 35 (100.0%) | **<0.001** | 0 (0.0%) | 19 (100.0%) | **<0.001** |
| High flow oxygen | 2 (1.4%) | 0 (0.0%) | 2 (5.7%) | 0.063 | 0 (0.0%) | 1 (5.3%) | 0.156 |
| Noninvasive ventilation | 34 (24.6%) | 0 (0.0%) | 34 (97.1%) | **<0.001** | 0 (0.0%) | 18 (94.7%) | **<0.001** |
| Invasive ventilation | 12 (8.7%) | 0 (0.0%) | 12 (34.3%) | **<0.001** | 0 (0.0%) | 7 (36.8%) | **<0.001** |
| **Clinical outcomes** | | | | | | | |
| From admission to critical illness | 2 (0–8) | — | 2 (0–8) | — | — | 5 (4–11) | — |
| Length of stay, days | 19 (17–21) | 19 (18–20) | 17 (15–19) | 0.766 | 19 (18–20) | 17 (14–20) | 0.575 |
| Length of stay in survivors, days | 20 (19–21) | 19 (18–20) | 22 (0–45) | **0.001** | 19 (18–20) | 22 (7–37) | 0.053 |
| Mortality | 25 (18.1%) | 0 (0.0%) | 25 (71.4%) | **<0.001** | 0 (0.0%) | 15 (78.9%) | **<0.001** |

Data are number (%), or median (95% CI).

[a] Patients progressed from severe type to critical ill type.

[b] Includes fluoroquinolones, β-lactams, macrolides, carbapenems, and glycopeptides.

1.763; P = 0.004) were associated with a higher risk, whereas higher lymphocyte count at admission (OR 0.147, 95% CI 0.028–0.760; P = 0.022) was associated with a lower risk of critical illness development (Table 4).

## Discussion

The mortality rate of COVID-19 patients in critical conditions remains high [9,10]. Identification of high-risk patients in advance may help improve outcome by providing more aggressive therapy. In the present study, we screened easily accessible factors that may early predict the development of or progression to critical illness. Of note, higher neutrophil count and lower lymphocyte count at admission were associated with an increased risk of critical illness in both all enrolled patients and those with severe illness at admission. Our results indicated that we should be alert to patients with these characteristics and consider measures to prevent disease progression.

Previous studies have revealed a significant difference in lymphocyte count between severe and non-severe cases [4,13], between intensive care unit (ICU) and non-ICU patients [4,6,14], as well as between survivors and non-survivors with COVID-19 [15]. Mechanisms leading to lymphopenia in COVID-19 patients are not clear but may include the following, i.e., lymphocyte death resulted from direct virus infection, lymphatic organs (such as thymus and spleen) damage due to direct virus infection, lymphocyte apoptosis induced by inflammatory cytokines, inhibition of lymphocytes by metabolic acidosis, and translocation of lymphocyte from peripheral blood to the target organs such as lungs [16,17]. Both lower lymphocyte count and lower lymphocyte percentage are strongly related to the severity of disease, they also predict

**Table 4. Predictors of progression to critically ill type.**

| Variables | Univariable analysis | | Multivariable analysis [a] | |
|---|---|---|---|---|
| | Odds ratio (95% CI) | P value | Odds ratio (95% CI) | P value |
| **All patients [b]** | | | | |
| Age, years | 1.035 (1.004–1.067) | 0.028 | —— | —— |
| Male gender | 2.196 (0.989–4.879) | 0.053 | —— | —— |
| Expectoration | 2.105 (0.896–4.945) | 0.087 | —— | —— |
| Smoking history | 2.328 (0.955–5.674) | 0.063 | —— | —— |
| From diagnosis to admission, days | 1.047 (0.992–1.106) | 0.098 | 1.108 (1.022–1.202) | 0.013 |
| Use of arbidol before admission | 0.188 (0.054–0.658) | 0.009 | —— | —— |
| Use of fluoroquinolones before admission | 0.385 (0.171–0.867) | 0.021 | —— | —— |
| Pulse oxygen saturation at admission <93% | 8.587 (3.605–20.456) | <0.001 | 5.775 (1.257–26.535) | 0.024 |
| Neutrophil count at admission, $\times 10^9$/L | 1.626 (1.359–1.946) | <0.001 | 1.495 (1.177–1.899) | 0.001 |
| Lymphocyte count at admission, $\times 10^9$/L | 0.199 (0.076–0.521) | 0.001 | 0.149 (0.026–0.852) | 0.032 |
| Albumin at admission, g/L | 0.859 (0.781–0.944) | 0.002 | —— | —— |
| Creatine kinase-MB, ng/mL | 2.662 (1.671–4.242) | <0.001 | 2.449 (1.089–5.511) | 0.030 |
| D-dimer at admission, μg/mL | 1.186 (1.090–1.290) | <0.001 | —— | —— |
| **Severe patients at admission [c]** | | | | |
| Age, years | 1.027 (0.990–1.065) | 0.161 | —— | —— |
| Male gender | 2.483 (0.876–7.037) | 0.087 | —— | —— |
| Expectoration | 1.800 (0.382–8.488) | 0.458 | —— | —— |
| Diabetes mellitus history | 2.421 (0.845–6.934) | 0.100 | —— | —— |
| From diagnosis to admission, days | 1.066 (1.001–1.136) | 0.048 | 1.085 (1.009–1.167) | 0.027 |
| Use of arbidol before admission | 0.229 (0.050–1.046) | 0.057 | —— | —— |
| Pulse oxygen saturation at admission <93% | 9.617 (3.132–29.528) | <0.001 | 11.182 (2.426–51.534) | 0.002 |
| Neutrophil count at admission, $\times 10^9$/L | 1.491 (1.222–1.820) | <0.001 | 1.403 (1.117–1.763) | 0.004 |
| Lymphocyte count at admission, $\times 10^9$/L | 0.302 (0.098–0.932) | 0.037 | 0.147 (0.028–0.760) | 0.022 |
| Creatine kinase-MB, ng/mL | 1.003 (1.001–1.006) | 0.012 | —— | —— |
| D-dimer at admission, μg/mL | 1.101 (0.998–1.215) | 0.054 | —— | —— |

[a] Factors with P <0.10 in univariable analyses or were considered clinically important were included in multivariable regression model with Wald (backward) method.

[b] Use of antiviral drugs was excluded due to collinearity and use of arbidol. Use of antibiotics was excluded due to collinearity with use of fluoroquinolones. White blood cell count was excluded due to collinearity with neutrophil count. Aspartate aminotransferase, blood urea nitrogen, lactate dehydrogenase, myoglobin and hypersensitive troponin I were excluded due to collinearity with creatine kinase-MB. Prothrombin time and activated partial thromboplastin time were excluded due to clinical correlation with d-dimer. Dyspnea, heart rate and respiratory rate were excluded due to clinical correlation with pulse oxygen saturation at admission <93%.

[c] White blood cell count was excluded due to collinearity with neutrophil count. Blood urea nitrogen, lactate dehydrogenase, and myoglobin were excluded due to collinearity with creatine kinase-MB. Prothrombin time and activated partial thromboplastin time were excluded due to clinical correlation with d-dimer.

the progression to critical illness. Tan et al. [16] established a time-lymphocyte percentage model and verified that lymphocyte percentage can reliably classify the severity (moderate, severe, and critical ill) of COVID-19 patients. In another study, Wang and colleagues reported that decrease of CD8+ T cells and B cells and increase of CD4+/CD8+ ratio were independently associated with poor outcomes [18]. In line with above results, we also found that low lymphocyte count at admission was an independent predictor of critical case or progression to critical illness.

Considering the significant decreases of lymphocyte count/percentage and their association with the severity and outcomes of COVID-19 patients, measures to increase lymphocytes may improve outcome. In previous studies, effective therapy was followed by increased lymphocytes [18,19]. Antiviral therapies are suggested for COVID-19 patients according to guidelines

issued by the Chinese National Health Commission [3]. In one study, early antiviral treatment was associated with less disease progression [11]. Thymosin, an immune enhancer by inducing T cell differentiation and maturity, is also used in some severe and critical COVID-19 patients but the efficacy remained unclear [18,20]. In the present study, critical cases of all enrolled patients received fewer antiviral drugs (especially arbidol) before admission, also indicating the potential effect of early antiviral therapy.

Neutrophil count and neutrophil percentage are significantly increased in severe and critically ill COVID-19 patients [13–15,21]. Our results also showed that high neutrophil count independently predicted the development of critical illness. Increased neutrophil suggests a possible bacterial infection or a non-infectious inflammatory response. In a study of 339 elderly patients with COVID-19, 143 cases (42.8%) had comorbid bacterial infection [22]. And the rate of bacterial co-infection was higher in severe/critical ill patients than in mild/moderate patients [11]. It is known that dry cough is the typical symptom of COVID-19 patients at early stage [4,6]. In the present study, 62.0% of patients had expectoration at admission, suggesting a potential bacterial co-infection in those severe cases; furthermore, patients who developed critical illness received less antibiotic therapy before admission. Therefore, active bacteriological surveillance, empirical antibiotic therapy in patients with suspected co-infection, and timely clarification of pathogenic bacteria may help to slow disease progression and improve outcome but requires demonstration. On the other hand, the study of Zhang and colleagues [13] excluded patients with common bacteria or viruses associated community-acquired pneumonia and those with procalcitonin level of greater than 0.5 ng/ml; they still found that neutrophil was higher in severe patients and those with poor outcomes (ICU admission, mechanical ventilation, or death). This indicates higher neutrophil count might also be a reflection of excessive inflammation.

A higher neutrophil count and a lower lymphocyte count, i.e., the increase of neutrophil-to-lymphocyte ratio, is demonstrated as a prognostic biomarker in cancer patients [23,24]. A meta-analysis including 6 studies of 824 patients found that a higher neutrophil-to-lymphocyte ratio predicted clinical severity and poor prognosis of patients with COVID-19 [25]. Our results are in line with the above studies and provide further clues that early combined antiviral and antibiotic therapy may be beneficial for COVID-19 patients. Further studies are required to confirm our hypothesis.

In the present study, long duration from diagnosis to admission was an independent predictor of disease progression. The surge of COVID-19 patients and the relative shortage of medical resources in early February of 2020 in Wuhan delayed the hospitalization of some patients. Failure to receive timely hospital treatment might be the main reason for disease progression. Similar phenomenon was also observed in the study of Wang et al. [6] that ICU patients had a longer duration from symptom onset to hospital admission when compared with non-ICU patients. Low oxygen saturation is related to the severity of lung injury and is an important indicator to initiate oxygen therapy and mechanical ventilation. Oxygen saturation <93% was one of the diagnostic criteria of severe COVID-19. Our results showed that pulse oxygen saturation of <93% at admission independently predicts the progression of the disease to critical illness. Therefore, care should be taken for these patients. We also found that higher level serum creatine kinase-MB predicted critical illness; similar result was reported by others [26,27].

There are several limitations in the present study. First, the sample size included in this study was relatively small. Larger sample size studies are required to verify our results. Second, we did not collect imaging examination (chest X-ray and CT scan) data because of the difficulty to quantify results. However, available evidences showed that CT results are significantly correlated with pulse oxygen saturation and lymphocyte numbers [28]; and that lymphocyte

percentage can be used reliably to classify disease severity without other auxiliary indicators [16]. Therefore, shortage of imagining results does not seem to change our results. Third, due to the retrospective nature of the study, bias may be introduced by unrecognized factors. Nonetheless, our results provide clues for interventional studies.

## Conclusions

In summary, results of this retrospective study showed that high neutrophil count and low lymphocyte count at admission were early independent predictors of progression to critical ill-ness in severe COVID-19 patients. The effects of early combined antiviral and antibiotic ther-apy on the outcomes of COVID-19 patients deserve further study.

## Supporting information

**S1 Table. Normal ranges of laboratory tests.**
(DOCX)

**S2 Table. Predictors of progression from severe type to critical type in COVID-19 patients (univariable logistic regression analysis).**
(DOCX)

**S1 Dataset. Relevant data underlying the main results.**
(XLSX)

## Acknowledgments

The authors gratefully acknowledge Dr. Hong-Yu Yang M.D. (Department of Nephrology, Peking University First Hospital, Beijing, China) for his help in the acquisition of data. We also thank all team members of Peking University in Wuhan for their great work in fighting against COVID-19.

## Author Contributions

**Conceptualization:** Nan Li, Hao Kong, Dong-Xin Wang, Hai-Chao Li, Xin-Min Liu.

**Data curation:** Nan Li, Xi-Zi Zheng.

**Formal analysis:** Nan Li, Xue-Ying Li, Dong-Xin Wang.

**Investigation:** Nan Li.

**Methodology:** Nan Li, Xue-Ying Li, Hai-Chao Li.

**Supervision:** Dong-Xin Wang, Hai-Chao Li, Xin-Min Liu.

**Visualization:** Hao Kong, Xi-Zi Zheng, Jing Ma, Hong Zhang, Dong-Xin Wang, Hai-Chao Li, Xin-Min Liu.

**Writing – original draft:** Nan Li, Hao Kong.

**Writing – review & editing:** Nan Li, Hao Kong, Jing Ma, Hong Zhang, Dong-Xin Wang, Hai-Chao Li.

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
