## [Decision Letter · Decision Letter 0]

11 Nov 2020

PONE-D-20-32906

Early predictive factors of progression from severe type to critical ill type in patients with Coronavirus Disease 2019: A retrospective cohort study

PLOS ONE

Dear Dr. Wang,

Thank you for submitting your manuscript to PLOS ONE. After careful consideration, we feel that it has merit but does not fully meet PLOS ONE’s publication criteria as it currently stands. Therefore, we invite you to submit a revised version of the manuscript that addresses the points raised during the review process.

The reviewers have commented on your above paper. They have suggested that this manuscript be revised according to the reviewers suggestions and resubmitted.  Substantially, the paper does have merit, but the language style needs to be improved in order to have a classic article. Provided you address the changes recommended, the manuscript will be accepted for publication. 

We look forward to receiving your revised manuscript.

Kind regards,

Prof. Raffaele Serra, M.D., Ph.D

Academic Editor

PLOS ONE

Additional Editor Comments:

The reviewers have commented on your above paper. They have suggested that this manuscript be revised for language polishing and resubmitted.

Journal Requirements:

2. In ethics statement in the manuscript and in the online submission form, please provide additional information about the patient records/samples used in your retrospective study. Specifically, please ensure that you have discussed whether all data were fully anonymized before you accessed them and/or whether the IRB or ethics committee specifically waived the requirement for informed consent.

Reviewers' comments:

Reviewer's Responses to Questions

**Comments to the Author**

1. Is the manuscript technically sound, and do the data support the conclusions?

Reviewer #1: Yes

Reviewer #2: Yes

2. Has the statistical analysis been performed appropriately and rigorously? 

Reviewer #1: Yes

Reviewer #2: Yes

3. Have the authors made all data underlying the findings in their manuscript fully available?

Reviewer #1: Yes

Reviewer #2: Yes

4. Is the manuscript presented in an intelligible fashion and written in standard English?

Reviewer #1: Yes

Reviewer #2: No

5. Review Comments to the Author

Reviewer #1: The authors aimed to identify factors that early predict the progression of COVID-19 from severe to critical illness. I think this is a timely and novel study. It is overall well written and fully detailed.

The authors should be congratulated for such a study. I support it for publication.

Reviewer #2: I really appreciate how this study is conducted and there are many important findings: neutrophil count and lower lymphocyte count at admission needs this way to be investigated in order to predict the risk in covid-19 patients.

Only the english language needs to be revised for style.

6. PLOS authors have the option to publish the peer review history of their article (what does this mean?). If published, this will include your full peer review and any attached files.

Reviewer #1: **Yes: **Nicola Ielapi

Reviewer #2: No

---

## [Author Response · Author response to Decision Letter 0]

16 Nov 2020

Journal Requirements:

Response: Thank you for reminding us. We have confirmed that our manuscript met PLOS ONE's style requirements, including those for file naming.

2. In ethics statement in the manuscript and in the online submission form, please provide additional information about the patient records/samples used in your retrospective study. Specifically, please ensure that you have discussed whether all data were fully anonymized before you accessed them and/or whether the IRB or ethics committee specifically waived the requirement for informed consent.

Response: Thank you. We have provided additional information about the patient records used in our retrospective study. “Considering that the study was retrospective in nature and no follow-up was performed, the Ethics Committee agreed to waive written informed consent. Personal data of patients were kept strictly confidential.” (page 6, lines 112-114).

3. We note that you have indicated that data from this study are available upon request. PLOS only allows data to be available upon request if there are legal or ethical restrictions on sharing data publicly. 

Response: Thank you. We have uploaded the minimal anonymized data set necessary to replicate our study findings as Supporting Information file (S3 Dataset: Relevant data underlying the main results.).

Reviewer #2:

1. The English language needs to be revised for style.

Response: Thank you. We have revised the English language style.

---

## [Editor Report · Decision Letter 1]

18 Nov 2020

Early predictive factors of progression from severe type to critical ill type in patients with Coronavirus Disease 2019: A retrospective cohort study

PONE-D-20-32906R1

Dear Dr. Wang,

We’re pleased to inform you that your manuscript has been judged scientifically suitable for publication and will be formally accepted for publication once it meets all outstanding technical requirements.

Kind regards,

Prof. Raffaele Serra, M.D., Ph.D

Academic Editor

PLOS ONE

Additional Editor Comments (optional):

amended manuscript is acceptable
---

## [Editor Report · Acceptance letter]

20 Nov 2020

PONE-D-20-32906R1 

Early predictive factors of progression from severe type to critical ill type in patients with Coronavirus Disease 2019: A retrospective cohort study 

Dear Dr. Wang:

I'm pleased to inform you that your manuscript has been deemed suitable for publication in PLOS ONE. Congratulations! Your manuscript is now with our production department. 

Kind regards, 

on behalf of

Prof. Raffaele Serra 

Academic Editor

PLOS ONE